# Pseudouridine Synthase RsuA Confers a Survival Advantage to Bacteria under Streptomycin Stress

**DOI:** 10.3390/antibiotics12091447

**Published:** 2023-09-14

**Authors:** Sudeshi M. Abedeera, Kumudie S. Jayalath, Jiale Xie, Rushdhi M. Rauff, Sanjaya C. Abeysirigunawardena

**Affiliations:** Department of Chemistry and Biochemistry, Kent State University, 1175 Risman Dr., Kent, OH 44242, USA; asudeshi@kent.edu (S.M.A.); kjayalat@kent.edu (K.S.J.); jxie10@kent.edu (J.X.); mmohame1@kent.edu (R.M.R.)

**Keywords:** ribosome, pseudouridine, pseudouridine synthase, streptomycin, helix18

## Abstract

Bacterial ribosome small subunit rRNA (16S rRNA) contains 11 nucleotide modifications scattered throughout all its domains. The 16S rRNA pseudouridylation enzyme, RsuA, which modifies U516, is a survival protein essential for bacterial survival under stress conditions. A comparison of the growth curves of wildtype and RsuA knock-out *E. coli* strains illustrates that RsuA renders a survival advantage to bacteria under streptomycin stress. The RsuA-dependent growth advantage for bacteria was found to be dependent on its pseudouridylation activity. In addition, the role of RsuA as a trans-acting factor during ribosome biogenesis may also play a role in bacterial growth under streptomycin stress. Furthermore, circular dichroism spectroscopy measurements and RNase footprinting studies have demonstrated that pseudouridine at position 516 influences helix 18 structure, folding, and streptomycin binding. This study exemplifies the importance of bacterial rRNA modification enzymes during environmental stress.

## 1. Introduction

Post-transcriptional ribonucleotide modifications are scattered in various biologically relevant RNAs in all three kingdoms of life. Currently, more than 170 post-transcriptional ribonucleotide modifications have been observed in nature. Although RNA nucleotide modifications are implicated in influencing local RNA structure and stability, RNA–RNA and RNA–protein interactions, and RNA folding and dynamics, the exact biological role of many RNA nucleotide modifications is unknown [1,2]. However, several known ribosomal RNA (rRNA) modifications are related to antibiotic resistance in bacteria. Additionally, several rRNA modification enzymes in bacteria are also correlated with antibiotic resistance [3,4,5,6,7]. There are 11 nucleotide modifications in bacterial small ribosomal subunit rRNA (16S rRNA) [1]. Out of the 10 nucleotide modification enzymes responsible for 16S rRNA modifications, functional mutants of KsgA and RsmG caused antibiotic resistance in bacteria [7,8]. In addition to their role in antibiotic resistance in bacteria, these two enzymes influence 30S ribosome subunit assembly. Interestingly, however, methyltransferase activity is not required for the role of KsgA as a ribosome assembly factor [6], suggesting a possible link between ribosome assembly and antibiotic resistance in bacteria.

Ribosomal small subunit pseudouridine synthase A (RsuA) is responsible for the single pseudouridine modification located at position 516 (*E. coli* numbering) of 16S helix 18 [9,10,11]. RsuA belongs to the RsuA family of pseudouridine synthases. It contains a core domain that carries the catalytic site and a peripheral domain that is required for its binding to rRNA [12]. Like many rRNA modification enzymes, RsuA preferably binds to a ribosome assembly intermediate [9]. The presence of ribosomal protein uS17 is advantageous for both RsuA binding and its activity [13]. Although RsuA is among many rRNA modification enzymes and ribosomal proteins classified as redundant proteins under normal growth conditions [10], it has recently been identified as a survival protein that plays a crucial role in the survival of bacteria under various environmental stress conditions [14]. Due to the ability of RsuA to bind closer to the streptomycin binding site and function as a “survival protein”, this study examines the capability of the RsuA protein to influence streptomycin tolerance in bacteria. In addition, the role of pseudouridine (Ψ516) in 16S helix 18 structure and streptomycin binding is also discussed.

## 2. Results

### 2.1. RsuA Provides a Survival Advantage to Bacteria under Streptomycin Stress

Streptomycin binds to a pocket in the 30S ribosome that is created by ribosomal protein uS12 and 16S helices 18, 27, and 44 (Figure 1) and interferes with its decoding function. Streptomycin-resistance mutations are found in 16S rRNA, ribosomal protein uS12, and rRNA methyltransferase enzyme RsmG [7,15,16,17,18,19,20]. However, streptomycin-resistance mutations are not reported in the RsuA that preferably binds to a 30S assembly intermediate with an extended helix 18 [13].

We hypothesized that bacteria are less resistant to streptomycin in the absence of RsuA, perhaps due to its role as a survival protein. To test this hypothesis, the growth of wildtype (Wt) and RsuA knock-out (ΔRsuA) strains of *E. coli* under streptomycin stress were compared. The growth of wildtype (Wt) and RsuA knock-out (ΔRsuA) strains of *E. coli* after 18 h of inoculation under varying streptomycin concentrations (0–200 µg/mL) was recorded. Approximately, a 2-fold decrease in the IC_50_ of streptomycin was observed for ΔRsuA *E. coli* strain (5.9 ± 1.3 µg/mL) compared with that for Wt *E. coli* (12.0 ± 1.2 µg/mL), suggesting that RsuA plays a supportive role in the survival of *E. coli* under streptomycin stress (Figure 2).

The study of bacterial growth kinetics can provide an idea of the mechanism by which RsuA provides a survival advantage to cells experiencing streptomycin stress. Bacterial growth curves were obtained at three different streptomycin concentrations based on the IC_50_ values for both Wt and ΔRsuA strains (Figure 3). Wt *E. coli* strain illustrates a significant drop in the log phase growth rates as the streptomycin concentration is increased (Appendix A). The asymptote of growth curves is also significantly decreased in response to increasing streptomycin concentrations (Figure 3a). Interestingly, the growth of Wt *E. coli* under 11 and 13.5 µg/mL streptomycin was recovered after ~600 min. However, in contrast, the growth curves obtained for the ΔRsuA *E. coli* strain did not demonstrate a second log phase at the same higher streptomycin concentrations (Figure 3b). The comparison of the growth curves for the Wt and ΔRsuA strains shows that the Wt *E. coli* strain possesses an ability to tolerate streptomycin that is lacking in the ΔRsuA strain, suggesting an ability on the part of RsuA to grant *E. coli* cells a survival advantage. The recovery of cell growth after 600 min (10 h) is likely due to the depleted effective concentration of streptomycin. However, the ΔRsuA strain cannot recover after 10 h. The existence of a subpopulation of bacterial cells that can survive streptomycin stress in the presence of protein RsuA is a plausible explanation for this observation. The decrease in the bacterial growth rates of the ΔRsuA strain compared with the Wt strain in the presence of streptomycin is perhaps due to the decreased translation rates of ribosomes lacking Ψ516. These observations suggest that the presence of RsuA renders a survival advantage for Wt *E. coli* under streptomycin stress, especially at higher streptomycin concentrations (11 and 13.5 µg/mL). 

RsuA was supplemented into the ΔRsuA *E. coli* strain using a protein overexpression plasmid to ensure that observed differences in bacterial growth are RsuA dependent. Wildtype RsuA overexpression plasmid (ΔRsuA + wtRsuA) was transformed into ΔRsuA *E. coli* cells. We then performed similar bacterial growth assays, as previously explained, under a 50 µM inducer (IPTG) concentration. The 50 µM IPTG concentration is the minimal inducer concentration at which an overexpression of RsuA is visible in an SDS-PAGE gel (Appendix A). Bacterial growth curves for the Wt *E. coli* strain were not significantly altered by adding 50 µM of IPTG (Appendix A). The initial short log phase and extended stationary phase in Wt and ΔRsuA *E. coli* strains were not observed in the RsuA-supplemented ΔRsuA *E. coli* (Figure 3a,c). Such differences may arise from the inability to control RsuA expression levels after induction with IPTG. This may be because bacterial ribosome biogenesis is negatively impacted in the presence of RsuA in high concentrations. The RsuA-dependent changes in bacterial growth may be due to either its pseudouridylation activity or may be a function of RsuA, and has yet to be discovered. For instance, RsuA may function as a ribosome assembly factor during antibiotic stress. To investigate the role of Ψ516 in streptomycin resistance, a functional mutant of RsuA was overexpressed in ΔRsuA *E. coli*. An overexpression plasmid encoding a catalytically inactive RsuA (ΔRsuA + mutRsuA) generated by site-directed mutagenesis was transformed into ΔRsuA *E. coli* [10]. The growth curves obtained at higher streptomycin concentrations (11 and 13.5 µg/mL) for ΔRsuA + mutRsuA *E. coli* showed a significant loss of bacterial growth at the end of 24 h, compared with ΔRsuA *E. coli* strain with wildtype RsuA being overexpressed (Figure 3c,d). Although the shape of the two growth curves for ΔRsuA + wtRsuA and ΔRsuA + mutRsuA *E. coli* strains looked similar at higher streptomycin concentrations, the growth rate after 600 min was found to be higher for ΔRsuA *E. coli* containing wtRsuA. This observation suggests that pseudouridylation at position 516 of 16S rRNA can influence the bacterial growth rate. 

### 2.2. RsuA Influences rRNA Maturation during 30S Ribosome Biogenesis

Several rRNA modification enzymes can also function as assembly factors in addition to their enzymatic activity. The absence of these enzymes causes ribosome assembly defects and leads to bacterial growth defects. The hypothesis that RsuA can influence bacterial ribosome assembly has been tested by comparing the ratios of 16S and 17S rRNA for wildtype and ΔRsuA *E. coli* strains in the presence and absence of streptomycin stress. Radiolabeled DNA oligonucleotide primers complementary and specific to 16S and 17S RNA were annealed to the total RNA extracted from these two strains. Native gel electrophoresis was performed to separate the rRNA–primer complexes from the free primer. Bands corresponding to rRNA–primer complexes were quantified, and 17S/16S fractions were calculated (Figure 4, Appendix A). During the early growth stages, the 17S composition was found to be very high for both wildtype and ΔRsuA *E. coli* strains (Figure 4). As the bacterial growth increased, the 17S/16S ratio decreased as expected. In addition, the ΔRsuA bacterial strain showed a higher 17S/16S ratio at 4 h and 8 h (Figure 4). Unfortunately, however, the 17S/16S ratio at 4 h and 8 h for bacteria under streptomycin stress was not determined due to the lower cell growth observed at the early stages. At the stationary phase, the ratio was similar for both wildtype and ΔRsuA *E. coli* strains under streptomycin stress, similar to what was observed with no streptomycin. 

### 2.3. The Ψ516 Modification Influences the h18 Structure

The presence of an extra hydrogen bond donor (N1H) and unique stacking properties give pseudouridine the ability to influence RNA structure and stability [20,21,22,23]. Many previous studies have illustrated pseudouridine-dependent structural changes in RNA upon changes in environmental conditions such as pH, Mg^2+^ concentration, and the presence of small molecules [24,25]. Such pseudouridine-induced changes in RNA structure and stability are sensitive to the location of the pseudouridine in the RNA sequence [26]. Ψ516 is located near the sharp bend of the 16S helix 18 formed in the pseudoknotted native helix 18 structure. The presence of several Mg^2+^ ions that are coordinated to helix 18 close to Ψ516 (Figure 5a) [27] has led to the hypothesis that pseudouridine plays a vital role in the folding of helix 18 to its native form with the addition of Mg^2+^. To test pseudouridine-dependent changes in helix 18 folding, circular dichroism (CD) experiments were performed for two helix 18 upper hairpin loop model RNAs with pseudouridine and uridine at position 516 (Figure 1a). The usage of model RNAs presents many advantages in studying the structure and stability of RNA. The modular nature of RNA structure can justify the use of short model RNAs in various biophysical studies. In addition, antibiotic binding to ribosomal RNA and many other RNAs, such as RRE RNA, is studied using model RNAs. 

The h18-Ψ model RNA comprises *E. coli* 16S helix 18 residues 511 through 540 with a pseudouridine at position 516, whereas the unmodified counterpart, h18-U model RNA, has the pseudouridine at position 516 replaced by uridine. CD spectra were obtained for both h18-Ψ and h18-U constructs at varying magnesium concentrations to track structural changes during helix 18 folding. Both of the helix 18 RNAs give rise to CD spectra typical for RNAs. In the absence of Mg^2+^ ions, both helix 18 model RNAs give rise to a dichroic maximum at approximately 266 nm in the CD spectra. However, the molar ellipticity value at the spectral peak at 266 nm for the modified helix 18 RNA is significantly higher than that for the unmodified helix 18 model RNA (Figure 5b). Such changes can arise from the changes in stacking interactions. As expected, subtle changes in the CD spectra have been observed with an increase in magnesium ion concentration, showing the ability of both RNAs to fold in the presence of magnesium (Figure 5c,d). For both helix 18 model RNAs used in this research, shoulders in the CD spectra are observed at ~260 nm, suggesting structural heterogeneity. Although the shapes of CD spectra for both the modified and the unmodified RNAs are found to be similar, contrasting Mg^2+^-dependent folding profiles were observed for the two RNAs. In the presence of the modification, the shoulder in the CD spectra (at 260 nm) of h18 RNA decreased as the Mg^2+^ was added up to 1 mM. However, when the Mg^2+^ concentration was increased beyond 1 mM, the shoulder started reappearing (Figure 5c,e). A similar folding profile was observed in all three repeats, despite the change in ellipticity at no Mg^2+^ for each repeat (Figure 5b). However, for the unmodified helix 18 model RNA, only a gradual decrease in the molar ellipticity was observed, and only at 260 nm (Figure 5d,f). These results suggest that the change in 16S helix 18 structure at increasing concentrations of Mg^2+^ is sensitive to the presence of pseudouridine modification at position 516.

RNase T1 footprinting was performed to identify regions of helix 18 that change with the addition of Mg^2+^, especially in a pseudouridine-dependent manner. RNase T1 cleaves the phosphodiester bond at the 5′-end of unpaired guanines in RNA. We carried out RNase T1 footprinting for both the h18-Ψ (Figure 6b) and h18-U RNA constructs at 0–25 mM Mg^2+^ concentrations (Figure 6a). At all Mg^2+^ concentrations tested, 5′-phosphodiester bonds of G521 and G524 located at helix 18 upper hairpin region were cleaved by RNase T1. The fraction of RNase cleavage at each guanine was obtained and compared with the intensity of the full-length or intact RNA band. The fraction of RNase cleavage was found to be higher for G524 compared with G521, indicating that G524 shows lesser tertiary interactions than that of G521 (Figure 6b–e). Surprisingly, however, G530, known to be unpaired and projected towards the helix 44 in the 30S X-ray crystal structure, and its neighboring guanine, G529, were protected from RNase cleavage (Figure 6a). Such deviation from the expected cleavage pattern could arise from the previously predicted alternate base pairing in the 530 loop [27,28,29]. Similar to CD spectrometric measurements, RNase T1 cleavage at G521 and G524 increased up to 1 mM Mg^2+^ and decreased for concentrations higher than 1 mM. In addition to RNase T1 cleavages at G524 and G521 observed in modified helix 18, the unmodified counterpart is also cleaved at G515 and G517 (Figure 6a). The ability of pseudouridine at 516 (Ψ516) to stabilize local structures may be the reason for the absence of RNase T1 cleavage at G515 and G517 for modified helix 18 RNA. Furthermore, the relative RNase cleavage for G524 and G521 was higher for unmodified helix 18, indicating a less structured helix 18 structure. The pattern of RNase cleavage at G524 was similar in both unmodified and pseudouridylated helix 18 model RNAs. The RNase cleavage at G515 slightly decreases with the addition of Mg^2+^, suggesting slight differences in structures of modified and unmodified model RNAs near the pseudouridylation site. Interestingly, the cleavage at G521 prompted an increase in unmodified helix 18 RNAs with increasing Mg^2+^ concentrations, whereas a decrease in metal ions may restore the stability that was absent due to the lack of modification (Figure 6g). However, a significant change in RNase T1 digestion at G517 was not observed as the concentration of Mg^2+^ was increased (Figure 6f).

### 2.4. Ψ516 Modification Increases Streptomycin Binding to Helix 18 

Streptomycin binds to a pocket formed by the 16S helices 1, 18, 27, and 44 and ribosomal small subunit binding protein uS12 (Figure 1b). The phosphate backbone residues G526 and G527 in the h18 upper hairpin loop form contacts with streptomycin [30,31]. With the ability of pseudouridine to influence helix 18 structure and folding, we hypothesized that streptomycin could bind to h18-Ψ and h18-U with varying affinity. CD spectroscopy was used to monitor any structural changes of helix 18 upon the addition of streptomycin to h18-Ψ and h18-U model RNAs. Furthermore, we used CD spectroscopic changes to determine binding affinities of streptomycin to each helix 18 model RNA. Streptomycin was titrated against h18-Ψ and h18-U constructs at pH 7.0 and 4 mM Mg^2+^ concentration. A redshift in the peak maximum for both RNAs was observed upon the addition of streptomycin (Figure 7a). At very high concentrations of streptomycin, the molar ellipticity decreased drastically, perhaps due to the non-specific binding of the antibiotic to RNA. The fraction of RNA bound to streptomycin was calculated using the change in molar ellipticity at 265 nm. The equilibrium dissociation constants (K_d_s) for streptomycin binding with h18-Ψ and h18-U constructs were found to be 23 ± 3 µM and 63 ± 1 µM, respectively (Figure 7b). 

The increased binding affinity of streptomycin to helix 18 RNA (3-fold) in the presence of pseudouridine modification at position 516 suggests that pseudouridine preferably stabilizes a structure of helix 18 that favors streptomycin binding. On the other hand, streptomycin may bind to two different regions of the helix depending on the presence of pseudouridine.

According to the X-ray crystal structure of the 30S ribosomal subunit bound to streptomycin, the antibiotic molecule interacts with C526 and G527 in the h18 upper hairpin [31]. RNase T1 footprinting was performed for h18-Ψ and h18-U RNAs in the presence of streptomycin to determine the regions of helix 18 that are affected by streptomycin binding (Appendix A). Unlike in the presence of increasing magnesium ions, exposure to RNase T1 increased with the addition of streptomycin for all four exposed guanines in unmodified helix 18 RNA (Figure 8c–f). A similar trend was also observed for the two exposed guanines in the modified helix 18 RNA (Figure 8a,b). However, the percentage change in exposure to RNase T1 for G521 and G524 in the presence of streptomycin slightly increased in the modified RNA compared with its unmodified counterpart. In addition, streptomycin was tightly bound close to G515 and G517 compared with G524 and G521 in the unmodified helix 18 RNA (Figure 8c–f). In the modified helix 18 RNA, streptomycin is bound tighter to G521 and G524 compared with the unmodified helix 18 RNA (Figure 8a,d, Appendix A). Similar to the previously observed X-ray crystallography data, these results indicate that streptomycin interacts with the helix 18 upper hairpin loop region only when Ψ516 modification is present. Although streptomycin interacts with the unmodified helix 18 RNA, those interactions are closer to the helix 18 lower stem region.

## 3. Discussion

In bacteria, the nucleotide modification process requires a set of unique modification enzymes [1]. The roles of many rRNA modification enzymes and their respective nucleotide modifications are yet to be discovered. Interestingly, bacteria can survive without many ribosome modification enzymes under optimal growth conditions [32]. In addition, ribosomes lacking nucleotide modifications can catalyze peptide formation [33]. Interestingly, however, several studies show that rRNA nucleotide modifications influence the efficiency of ribosome assembly and its peptidyltransferase activity [33,34,35].

This research clearly illustrates a growth disadvantage for *E. coli* cells lacking RsuA under streptomycin stress. Similar to previous bacterial growth studies for RsuA deletion strains, no significant growth defects were observed in the absence of streptomycin [36]. The increased lag time for the growth of persistent bacteria in the presence of streptomycin suggests that bacteria are able to build mechanisms by which to tolerate streptomycin [37]. However, the lack of RsuA may influence these tolerance mechanisms. The increase in the growth lag in bacterial strains with RsuA present is likely due to the slow production of specialized ribosomes capable of fighting antibiotic stress, as suggested previously by Gorini et al. [38]. It is also possible that the production of fully assembled ribosomes is stalled in bacteria, perhaps due to the decreased ribosomal protein pool. In addition, the existing 70S ribosomes may operate with an altered translation rate. Bacteria lacking RsuA resulted in slow growth rates, and the asymptote at the stationary phase decreased with the increasing concentration of streptomycin. These observations suggest that bacteria lacking RsuA may synthesize ribosomes with reduced translation elongation rates [39]. Interestingly, our RNase T1 footprinting assays illustrate that the Ψ516 changes the confirmation of G530, which is known to influence mRNA decoding, also suggesting the ability of pseudouridylation at position 516 to influence translation rates. In addition, the lack of pseudouridylation can decrease the translation rates of streptomycin-bound 70S ribosomes even further compared with its presence, as is evident from the slow growth rates in the exponential growth phase. The universally conserved A1492 and A1493 of 16S helix 44 and G530 of 16S helix 18 undergo conformational changes during the decoding process [40]. The inability of G530 to undergo conformational changes due to streptomycin binding at the tip of helix 18 leads to translation defects. Our data illustrates that the lack of pseudouridylation can cause structural changes in the 530 loop. In addition, the stability of helix 18 and its flexibility may also decrease in the absence of pseudouridylation. However, the binding of streptomycin to helix 18 lacking the pseudouridylation may trap helix 18 in a non-native conformation that disfavors ribosome function. Therefore, the structural changes caused by the lack of pseudouridine may generate ribosomes that are less tolerant to streptomycin. In addition to changes in translation rate, pseudouridylation and the presence of RsuA can influence ribosome biogenesis. The increase in 17S observed in this research may be correlated to the deficiencies in ribosome assembly and 30S assembly intermediates produced in the absence of RsuA may have a weaker tendency to be matured. Previous studies have shown the ability of the RsuA to influence the binding of S17 and, hence, influence the binding of late-binding proteins such as S12. 

Streptomycin interacts with the 530 loop of 16S helix 18 [31]. Interestingly, streptomycin binding does not cause structural changes in helix 18 in fully modified native ribosomes [31]. However, the pioneering work undertaken by Powers and Noller illustrates that ribosomes with different helix 18 structures have contrasting affinities to streptomycin [41]. It is also likely that the local structural changes caused by the pseudouridine modification can alter the binding orientation of streptomycin to helix 18. Our study shows that the presence of pseudouridine at position 516 changes the stability of streptomycin–helix 18 complexes. Unexpectedly, streptomycin binds to helix 18 model RNA with pseudouridine tighter than the unmodified counterpart. The change in the streptomycin affinity to unmodified helix 18 model RNA compared with its modified counterpart may have resulted from the binding of streptomycin to two different regions of the two helix 18 model RNAs. RNase T1 footprinting experiments confirm the contrasting streptomycin binding modes for both model RNAs. However, our experiments with model RNAs may not sufficiently represent the binding affinity of streptomycin to 70S ribosomes due to the lack of interactions with the neighboring helices and protein uS12. The slower growth rates for the ΔRsuA *E. coli* strain supplemented with the RsuA functional mutant compared with wildtype *E. coli* strain also supports the idea that the pseudouridylation function of RsuA is also critical for the streptomycin resistance in bacteria. Interestingly, in *M. tuberculosis*, A514C and C517U mutations located in 16S helix 18 show resistance towards streptomycin, whereas, in *T. thermophilus*, C507A and G524U mutants are found to be streptomycin-dependent mutants [42,43,44]. All of these mutations can influence helix 18 pseudoknot stability. Furthermore, both the RsmG and uS12 that bind to the pseudoknotted helix 18 also carry streptomycin-resistance mutations. Similarly, the absence of pseudouridylation at position 516 likely influences the pseudoknot formation. Pseudouridylation at position 516 can influence the accurate positioning of G530 at the 30S decoding center by folding helix 18 to its pseudoknotted native structure. On the other hand, the lack of Ψ516 can also influence the accurate binding of ribosomal protein uS12 and modification enzyme RsmG that binds to pseudoknotted helix 18 [27,45] and carries streptomycin-resistance mutations. Any defects in the binding of uS12 and RsmG, and the lack of RsmG methyltransferase activity may also lead to slow translation rates. 

Some growth defects observed in *E. coli* lacking RsuA were absent when catalytically inactive RsuA is overexpressed, indicating that the RsuA enzyme has a unique role in streptomycin resistance in addition to Ψ516. It is likely that the lack of nucleotide modification and their respective modification enzymes produce unique sub-optimally active ribosomes that are not sturdy enough to survive under various cellular stress conditions. Different subpopulations of ribosomes can exist in different stages of growth as well as various environmental stress conditions [14,46,47,48,49]. MazEF is the most studied toxin–antitoxin (TA) system in *E. coli* that is triggered under stress conditions. MazEF-based cell death is considered a population phenomenon under stress conditions where it still results in the survival of a small subpopulation of cells. MazF leads to the selective synthesis of about 10% of total cellular proteins. The majority of the population has the expression of “death proteins”, while a small subpopulation will have “survival proteins” expressed [14]. Interestingly, RsuA is one of the survival proteins [14,47]. An increase in the bacterial growth lag in the presence of inactive protein similar to that of the wildtype strain suggests that the presence of the RsuA protein is equally important for streptomycin resistance as its pseudouridylase activity. RsuA may stabilize the binding of uS17 and hence influence the binding kinetics of late-binding proteins such as uS5 and uS12 [13]. The absence of RsuA through its alleged role as an assembly factor can deplete the pool of newly assembled ribosomes by slowing down the assembly process.

In summary, pseudouridylation of *E. coli* 16S rRNA at position 516 is likely to be critical for the structural integrity of helix 18 and its dynamics during mRNA decoding and may significantly influence bacterial growth in the presence of antibiotics that bind to its vicinity. In addition, it is also likely that the presence of RsuA, regardless of its activity, can influence the tolerance against streptomycin.

## 4. Materials and Methods

### 4.1. Experimental Details Preparation of Helix 18 RNA

This study used two chemically synthesized 30 nt long 16S helix 18 (residues 511–540; *E. coli* numbering) model RNAs. The h18-Ψ model RNA contained a pseudouridine residue at position 516, whereas the h18-U model RNA contained uridine at position 516. These two model RNA oligonucleotides were purchased from Horizon Discovery in the 2′-ACE-protected form. RNAs were deprotected by incubating them for 30 min at 60 °C in a 2′-deprotection buffer (100 mM acetic acid pH 3.4–3.8, adjusted with TEMED) provided by Horizon Discovery. Deprotected model RNAs were then dried in vacufuge and stored at −20 °C until further use. RNA stocks for CD experiments were made by dissolving the deprotected aliquots in TE (1 M Tris–HCl pH 7.5 and 0.5 M EDTA pH 8.0) buffer. The concentrations of RNA stock solutions were calculated using A_260_ absorbance and appropriate extinction coefficients provided by the manufacturer. RNA samples used for RNase T1 footprinting experiments were radiolabeled at the 5′-end with ^32^P isotope using standard T4-polynucleotide kinase (PNK)-based end-labeling protocols. The radiolabeled RNAs were then purified using 16% denaturing polyacrylamide gel. RNA samples were electrophoresed for 1 h at 15W per gel. An autoradiograph was developed to identify the band representing the full-length RNA. The identified full-length RNA band was exercised, and the radiolabeled RNA was extracted onto 1 mL of TEN buffer (10 mM Tris–HCl at pH 7.5, 1 mM Na_2_EDTA at pH 8.0, 250 mM NaCl) using the freeze–thaw method. Extracted RNAs were precipitated and dried before use.

### 4.2. Circular Dichroism (CD) Spectroscopy

CD spectra (200–320 nm) for model RNAs were obtained using a JASCO J-810 circular dichroism spectropolarimeter equipped with a water bath to control the temperature. Model RNA samples (5 μM) were resuspended in 500 µL of CD buffer (50 mM KCl, 20 mM sodium cacodylate, 0.5 mM EDTA, pH 7.6 adjusted with 1 M HCl). Three scans performed at 30 °C were averaged for each Mg^2+^ (0–25 mM) or streptomycin concentration (0–0.5 mM). The averaged CD spectra were then smoothed using the Savitsky–Golay algorithm (*n* = 15). All CD experiments were performed in triplicate to ensure reproducibility. The fraction of model RNAs bound to streptomycin was calculated using CD at 265 nm and plotted against streptomycin concentrations. Equilibrium dissociation constants (K_d_s) for streptomycin–RNA complexes were obtained by least-square fitting of the fraction-bound versus streptomycin concentration curves of the binding isotherms, using the Origin program. 

### 4.3. RNase Footprinting Experiments 

RNase T1 or RNase A footprints for the h18-Ψ and h18-U RNAs were obtained at various Mg^2+^ concentrations (0 mM–25 mM) or streptomycin concentrations (0 mM–0.5 mM). Each RNase T1/A digestion reaction was carried out by incubating 15 pmols of h18-Ψ/U RNAs and 1 μg of *E. coli* tRNA with 0.02 U of enzyme for 5 min at room temperature. Magnesium or streptomycin concentrations were adjusted with MgCl_2_ stock solutions (0–50 mM) or streptomycin sulfate stock solutions (0–2.5 mM), respectively. Digestion reactions were stopped by adding 1 mM aurin tricarboxylic acid and an equal volume of buffered formamide. Digestion products were run on a 16% polyacrylamide sequencing gel at 55W for 45 min. An autoradiograph was developed, and the intensities of RNase T1 cuts were quantified using the ImageJ software. The intensity of each band was normalized for the total intensity of the lane. The relative streptomycin cleavage versus streptomycin concentration curves was fitted to the binding isotherm as described in the previous section to obtain K_d_ for individual interaction.

### 4.4. Growth Inhibition Assays

The RsuA knock-out (ΔRsuA) strain and the respective wildtype strain of *E. coli* (*E. coli* K-12 BW25113: rrnB3 ΔlacZ4787 hsdR514 Δ(araBAD)567 Δ(rhaBAD)568 rph-1) were treated with varying concentrations of streptomycin ranging from 0–200 µg/mL in LB medium. They were grown at 37 °C for 18 h, and absorbance (at 600 nm) was measured with a SpectraMax4 spectrometer. The data were obtained in independent biological triplicates, and the average of the absorbance values (normalized) was plotted against (streptomycin) (µg/mL) and fitted with Origin to Equation 1 to determine the IC_50_ value.

### 4.5. Generation of Recombinant pCA24N Vector Encoding Catalytically Inactive RsuA 

The recombinant pCA24N vector containing the coding sequence for wildtype RsuA was purchased from the ASKA collection [50]. The RsuA functional mutant (D102N) was generated by site-directed mutagenesis using a Q5 site-directed mutagenesis kit purchased from NEB. The forward primer (5′-GGGGCGGTTGAATATTGATACC-3′) and reverse primer (5′-GCCGCATGCAGT TTCCAC-3′) were purchased from IDT. The mutated plasmids were transformed first into *E. coli* DH5α competent cells. Plasmid DNA was extracted using QIAprep Spin Miniprep Kit (Qiagen Inc., USA) following the manufacturer-suggested protocol. The mutation was confirmed by DNA sequencing (Eurofins Genomics Inc., USA). 

### 4.6. Generation of ΔRsuA + wtRsuA and ΔRsuA + mutRsuA E. coli Strains 

Competent cells from the *E. coli* strain ΔRsuA were prepared and transformed with recombinant pCA24N vector containing the coding sequence for wildtype RsuA (wtRsuA) and catalytically inactive RsuA (mutRsuA) to generate the ΔRsuA + wtRsuA, and ΔRsuA + mutRsuA *E. coli* strains, respectively. Transformed cells were then plated on LB agar plates containing 50 µg/mL chloramphenicol and incubated overnight at 37 °C. A single colony from each plate was used to prepare a glycerol stock of ΔRsuA + wtRsuA and ΔRsuA + mutRsuA *E. coli*. The expression of wtRsuA and mutRsuA (28 kDa) in *E. coli* (ΔRsuA) was confirmed by overexpression in the presence of 0–1 mM IPTG for 8 h followed by SDS-PAGE. Protein bands in the SDS-PAGE gels were visualized using coomassie staining. The overexpressed RsuA band intensity increased with the IPTG concentration. These gels were used to determine the IPTG concentration that gives the expected minimum level of RsuA overexpression. The selected IPTG concentration was used to obtain the growth curves for ΔRsuA + wtRsuA, and ΔRsuA + mutRsuA *E. coli* strains at varying streptomycin concentrations. 

### 4.7. Growth Curve Analysis

The growth curves for wildtype (Wt), ΔRsuA, ΔRsuA + wtRsuA, and ΔRsuA + mutRsuA *E. coli* strains were obtained at 4 selected streptomycin concentrations from 0, 6, 11, and 13.5 µg/mL. Bacterial cultures from respective *E. coli* strains were started as a 100X dilution from an overnight culture in fresh LB media and were grown at 37 °C. The absorbance was recorded at different time intervals up to 24 h with a SpectraMax4 spectrometer. All experiments were undertaken in triplicate, and the average absorbance values for each streptomycin concentration were plotted against time (minutes). IPTG was added to the growth medium to a final concentration of 50 µM to obtain growth curves of the ΔRsuA + wtRsuA and ΔRsuA + mutRsuA *E. coli* strains.

### 4.8. Non-Denaturing Gel Assays

Bacterial pellets were collected from wildtype (Wt) and ΔRsuA strains of bacteria in the presence and absence of streptomycin (13.5 µg/mL) at 4, 8, 15, and 24 h of growth. Total RNA was extracted from each bacterial pellet using TRIzol Max Bacterial RNA Isolation Kit (Ambion Inc., Austin, TX, USA). Two DNA primers (5′-AGTCTGGACCGTGTCTC-3′, 5′-GAA TTAAACTTCGTAATGAATTAC-3′) that bind to mature 16S rRNA sequence and 17S leader sequence, respectively, were designed and purchased from Integrated DNA Technologies Inc. The ^32^P-labelled DNA primer (1.5 pmol) was mixed with total RNA (2 µg) in HK buffer (80 mM HEPES, 330 mM KCl), and the primer was annealed to rRNA by heating at 50 °C for 1 h and slow-cooling to room temperature. The samples were mixed with the non-denaturing loading buffer and run in an 8% non-denaturing PAGE gel. The intensities of the gel bands after phosphorimaging were measured to obtain the relative amounts of total rRNA (16S + 17S rRNA), 17S rRNA, and 16S rRNA (total RNA—17S rRNA). The total RNA, 17S rRNA, and 16S rRNA were plotted to investigate the influence of RsuA deletion for rRNA processing and, thereby, ribosome assembly.

## 5. Conclusions

Pseudouridylation enzyme RsuA and its activity can influence the growth of bacteria under streptomycin stress. The ability of Ψ516 to influence 16S helix 18 folding and streptomycin binding is partly the cause of streptomycin tolerance in bacteria. In addition, RsuA may function as a ribosome assembly factor that plays a role during streptomycin stress that is yet to be discovered. 

## Figures and Tables

**Figure 1 antibiotics-12-01447-f001:**
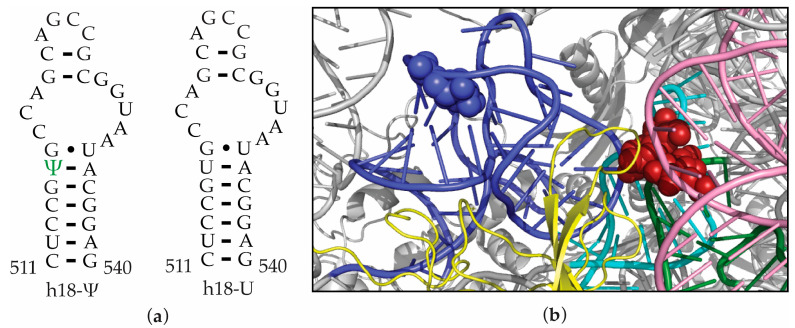
Streptomycin binds close to 16S helix 18. (**a**) 16S helix 18 model RNAs used in this research are shown. (**b**) Streptomycin binds to the pocket formed by ribosomal protein uS12 (yellow) and 16S helices 1 (green), 18 (blue), 27 (cyan), and 44 (pink) as observed in the X-ray crystal structure of streptomycin-bound 30S ribosome (PDB ID 4V50). Blue and red spheres represent Ψ516 and streptomycin, respectively.

**Figure 2 antibiotics-12-01447-f002:**
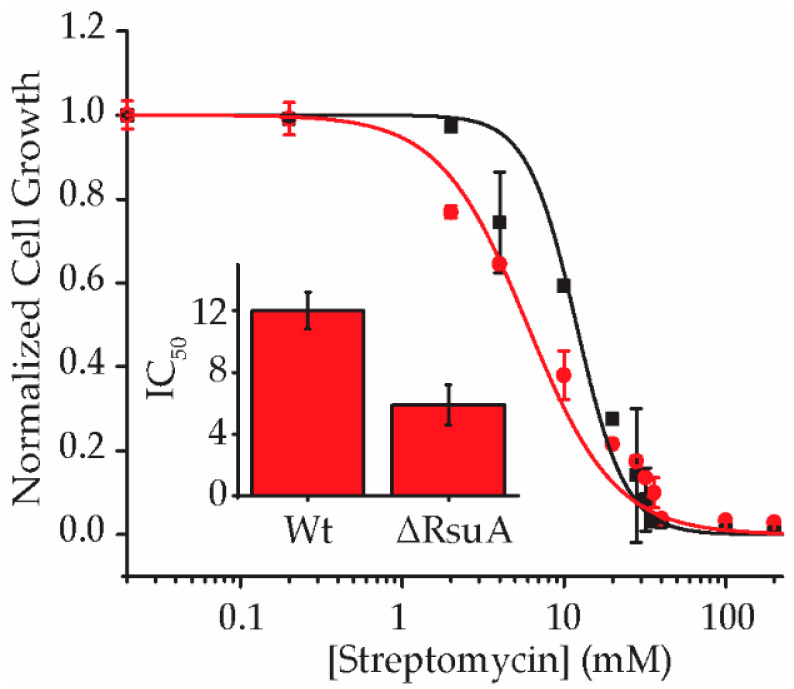
RsuA increases the resistance toward streptomycin. The normalized cell growth for Wt (black squares) and ΔRsuA (red squares) *E. coli* strains are plotted at varying streptomycin concentrations (0–200 μg/mL). The inset shows the corresponding IC_50_ values (μg/mL) for Wt and ΔRsuA *E. coli* strains.

**Figure 3 antibiotics-12-01447-f003:**
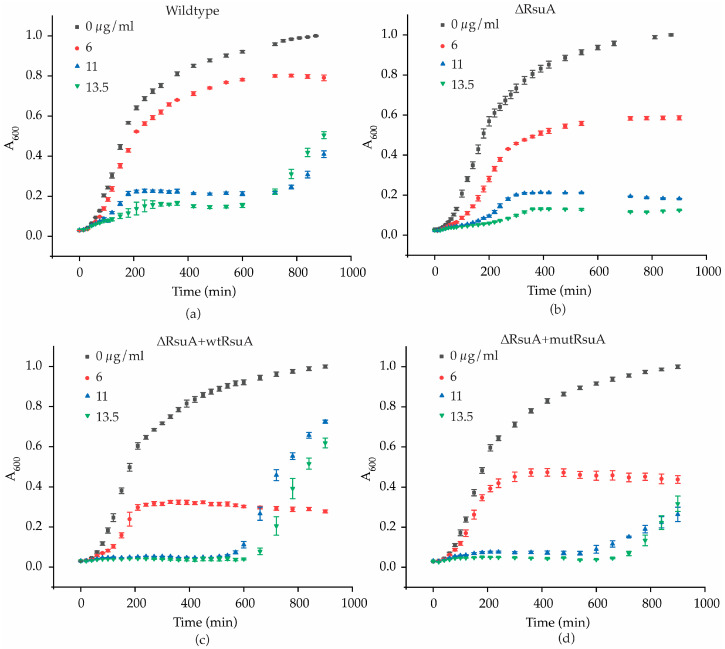
RsuA influences bacterial growth kinetics during streptomycin stress. Bacterial growth curves obtained for (**a**) wildtype (Wt) *E. coli* and (**b**) RsuA knock-out (ΔRsuA) strain of *E. coli*, and RsuA knock-out strain expressing (**c**) wildtype RsuA (ΔRsuA + wtRsuA) and (**d**) mutant RsuA (ΔRsuA + mutRsuA) in the background (50 μM IPTG), at varying streptomycin concentrations from 0–13.5 μg/mL, are shown. The average of the biological triplicates is shown.

**Figure 4 antibiotics-12-01447-f004:**
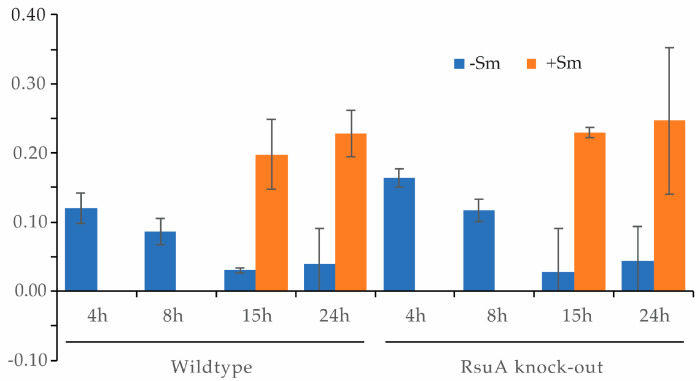
The 17S/16S ratio for wildtype (Wt) and RsuA knock-out (ΔRsuA) strains of *E. coli* in the presence and absence of streptomycin (Sm) stress. Error bars represent the SD of triplicates.

**Figure 5 antibiotics-12-01447-f005:**
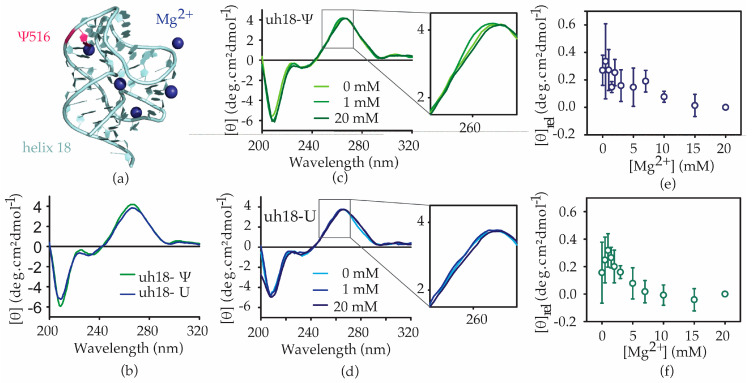
The Ψ516 modification influences the folding of helix 18. (**a**) The 16S helix 18 in the 30S X-ray crystal structure (PDB ID: 4V50) is shown. Blue spheres represent Mg^2+^ present near helix 18. Ψ516 is shown in magenta. (**b**) A comparison of circular dichroism (CD) spectra for h18-Ψ (green) and h18-U (blue) model RNAs. CD spectra of (**c**) h18-Ψ and (**d**) h18-U at various magnesium concentrations are shown. The inset shows the change in the 250–270 nm wavelength region of each spectrum. Changes in molar ellipticity with [Mg^2+^] for (**e**) h18-Ψ and (**f**) h18-U are shown. Standard deviation of triplicates is shown as error bars.

**Figure 6 antibiotics-12-01447-f006:**
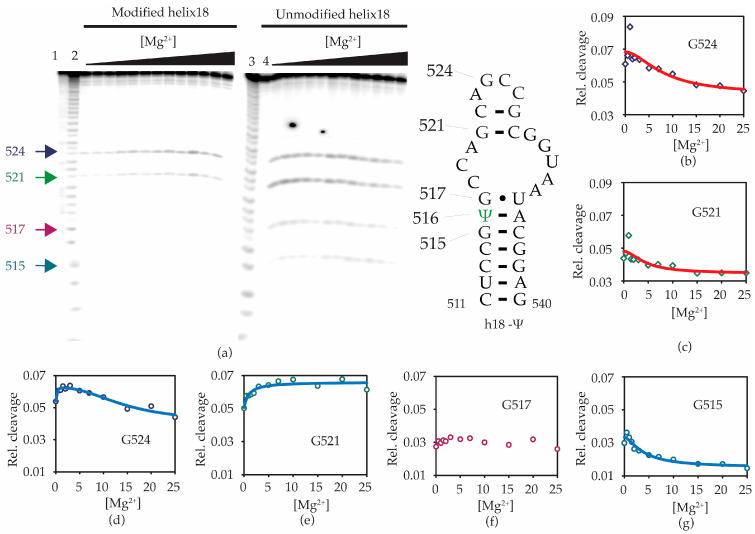
Ψ516 modification changes local RNA structure. (**a**) Radiograph showing RNase T1 digestion pattern of h18-Ψ (left panel) and h18-U (right panel) model RNAs. Bands corresponding to RNase T1 cleavages at G524, G521, G517, and G515 are shown in blue, green, purple, and teal arrows, respectively. The relative cleavage at each guanine of h18-Ψ (**b**,**c**) and h18-U (**d**–**g**) compared with full-length intact RNA at various Mg^2+^ concentrations (mM) are plotted.

**Figure 7 antibiotics-12-01447-f007:**
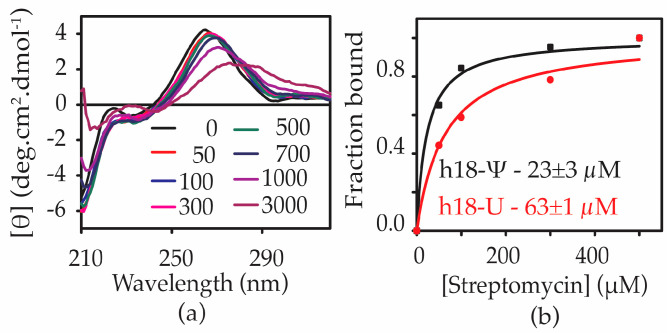
Ψ516 modification increases the affinity of streptomycin to helix 18. (**a**) CD spectra obtained for the h18-Ψ model RNA, at various streptomycin concentrations (0–3 mM) are shown. Experiments were performed at 30 °C in CD buffer (20 mM potassium cacodylate pH 7.0, 15 mM KCl, 4 mM MgCl_2_). (**b**) The fraction of RNA complexed with streptomycin at each streptomycin concentration is shown. The dissociation constants (K_d_s) of streptomycin binding for h18-Ψ and h18-U model RNAs were obtained by least-square fitting of binding curves to binding isotherm. All the experiments were undertaken in triplicate to confirm the reproducibility.

**Figure 8 antibiotics-12-01447-f008:**
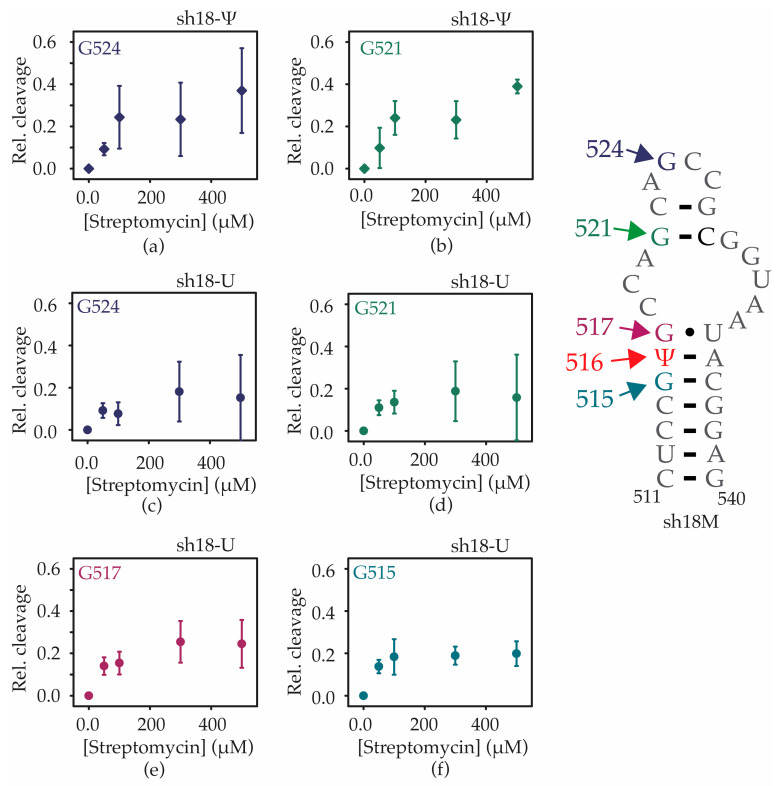
Streptomycin interacts with different regions of the h18 model RNAs with contrasting affinities. Relative cleavage at (**a**) G524 and (**b**) G521 of h18-Ψ and (**c**) G524, (**d**) G521, (**e**) G517, and (**f**) G515 of h18-U model RNAs at various concentrations of streptomycin are shown. Error bars shown on graphs represent the standard deviation of three replicates.

## Data Availability

Data is available upon request.

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
