# Peer review of "Pseudouridine Synthase RsuA Confers a Survival Advantage to Bacteria under Streptomycin Stress"

_antibiotics, 2023, doi:10.3390/antibiotics12091447_

Round 1

Reviewer 1 Report

RsuA is responsible for pseudouridine synthesis at position 516 of in helix 18 of 16S rRNA. In the manuscript entitled "Pseudouridine synthase RsuA confers a survival advantage to bacteria under streptomycin stress", the authors examine the ability of RsuA to enhance survival under stress induced by the aminoglycoside antibiotic streptomycin. Streptomycin binds to the 30S ribosomal subunit in close proximity to U516 and mutations in 16S rRNA or ribosomal protein uS12 confer streptomycin resistance.

The growth curves in Figure 3 are presented as linear plots. These are traditionally presented as semi-log plots. Is there some reason for using linear plots? Also, growth curves should be presented with error bars indicating the standard deviation, rather than just showing the average values. Also, the colors of the data points are similar enough to make it difficult to read. Please either use different symbols to represent the data points or use more easily distinguishable colors. Based on these data, the authors should provide a table showing the log phase growth rate of each of these strains.

Also shown in the growth curves in Figure 3, at higher streptomycin concentrations (11 and 13.5 ug/ml), there is growth inhibition until around 800 minutes, after which there is robust growth. What is this growth attributed to? Given the low concentrations of streptomycin used, the authors may want to consider that these are the result of spontaneous deletions of rsmG, encoding a 16S rRNA methyltransferase, and which have been shown to confer weak streptomycin resistance. If true, such deletions would appear at random times during the experiment, and the timing of these increases in growth should be highly variable. Unfortunately, because only the averages are shown, it is difficult to determine if such high variability was observed. The authors draw the conclusion that this represents a recovery of the population after prolonged incubation, but do not show how reproducible this phenomenon is. Showing the standard deviations will greatly help their case. The authors should also test cells at the end of the experiment to see if they are still streptomycin sensitive or if they have acquired resistance (either an rsmG mutation or an rpsL mutation).

The authors then examine the effect of the RsuA knockout on the 17S/16S ratio, as an indicator of ribosome assembly. They find this ratio decreases as cultures move from log phase to stationary phase, and that the ratio is higher for the RsuA knockout strain at earlier time points. However, no interpretation is given. The data for stationary phase under streptomycin stress are unfortunately, in this reviewer's opinion, too variable to allow any conclusion to be drawn.

The authors then use CD spectroscopy to examine the Mg-dependence of folding of model h18 hairpin loops with and without pseudouridine. This seems like reasonable experiment to perform; however, this reviewer does not have the expertise to evaluate the significance of the differences the authors describe.

The authors perform structure probing of their h18 model RNAs using RNase T1. They show that lack of pseudouridylation leads to the appearance of new cleavages after G515 and G517 as well as substantial increases in cleavage after G521 and G530. The latter is significant due the role of G530 in the decoding of mRNA as shown by the Ramakrishnan group. The authors seem to neglect this point, and it would improve the manuscript if they were to include some discussion of the implications of this finding. The authors further use RNase T1 to footprint streptomycin onto their model RNAs. 

One major concern about the model RNAs used in this study is the complete absence of the internal loop needed for formation of the pseudoknot structure that is a central feature of 16S rRNA helix 18. This pseudoknot involves positions 505-507 forming Watson-Crick pairs with positions 524-526. The loop consisting of 505-507 is absent, and so the model RNA cannot possibly form the pseudoknot structure. This raises serious questions about the structures of these model RNAs and their relevance to the binding of streptomycin. Another criticism of the model RNAs is that they lack the m7G527 modification produced by RsmG. This is important because lack of this modification confers weak streptomycin resistance. The authors need to point this out as a caveat to their experiments. Based on their footprinting, the authors make some conclusions about the binding contacts with streptomycin, but do not sufficiently clearly compare their observations with crystal structures of streptomycin bound to the ribosome. This is an important validation point.

In summary, this manuscript describes an interesting study on the role of pseudouridylation of 16S rRNA helix 18. The paper is well written and clearly and logically presented. However, there are deficiencies in some of the experiments that demand substantial revision. The authors also need to be more cautious in their concluding paragraph in which they state that "Pseudouridylation ... is likely to be critical for the structural integrity of helix 18 and its dynamics during mRNA decoding ..." This is something of an overstatement, given that the pseudouridylation is not essential for growth, and the authors have not at all addressed the question of mRNA decoding. What they can say is that pseudouridylation influences local RNA conformation and that their experiments with model RNAs, though not ideal models, do suggest that it impacts binding of streptomycin.

Minor points.

1. The authors repeatedly use the phrase "streptomycin-resistant mutations". This should read "streptomycin-resistance mutations". Mutations confer resistance, mutants are resistant.

2. "Streptomycin" in the last sentence of the Figure 3 legend should not be capitalized.

3. On line 146, the authors refer to Figure 4b. However, Figure 4 is not divided into panels.

The quality of the writing is sound. Some minor typos found.

Reviewer 2 Report

The paper entitled " Pseudouridine synthase RsuA confers a survival advantage to bacteria under streptomycin test"  is a very interesting study in the field of biocmemistry. This progect is of great scientific soundness, broadening our knowledge about ribosome acting antibiotics. I suggest it should be accepted aftter minor revisions.

1. Authors should add author contribution to the relevant section

2. In my opinion it is not clear enough to all readers why certain concentrations of streptomycin are chosen. Also, in fig 3c it is interesting why there is no restore of cell growth at the concentration of 6 μg/ml contrary to bigger ones.Can it be explained??

3. In the line 147 in section 2.2 it is mentioned that the ratio 17/16S is not measured because of no growth. How is it possible to be measured afterwards?I think it is confusing.

Reviewer 3 Report

The present study dissects the role of a bacterial RNA pseudouridine synthase, RsuA, and its directed pseudouridylation (ψ) under streptomycin stress. RsuA converts U516 of bacterial 16S ribosomal RNA (rRNA) to ψ516, located at helix 18 (h18) of 16S rRNA, which forms part of the binding pocket of the antibiotic streptomycin. The authors found that RsuA confers a survival advantage for wild-type (WT) E. coli under streptomycin stress, especially at higher streptomycin concentrations. They also provide evidence of an rRNA maturation defect in ΔRsuA at early stages, independent of streptomycin. Circular dichroism (CD) spectroscopy and RNase T1 footprinting assays demonstrated that ψ516 modification influences the h18 structure, which is related to Mg2+. Lastly, the authors attempted to test whether ψ516 modification affects streptomycin binding affinity to h18 using CD spectroscopy and RNase T1 assays, but the data and interpretations are somewhat confusing.

The findings from this work are interesting and will enhance our knowledge regarding the role of RNA modification enzymes and their modifications, which are under-studied. However, several key weaknesses are evident in this study. Firstly, the growth curves of ΔRsuA supplemented with WT RsuA plasmid or RsuA mutant plasmid are interesting but somewhat confusing. The ΔRsuA + WT RsuA plasmid cells do not behave like WT cells, and the authors attribute this to uncontrolled RsuA levels. This also raises questions about the conclusions drawn from the RsuA mutant experiments, as it appears that at 6ug/ml streptomycin, ΔRsuA cells supplemented with mutant RsuA grow better than those supplemented with WT RsuA plasmid. Did the authors try WT cells supplemented with WT RsuA plasmid? If overexpressing RsuA from the plasmid is causing abnormal growth curves, it should be observed in such cells. A related question to this point is, do the authors know the stoichiometry of ψ516 in WT E. coli? This seems to be a key missing piece of information that could impact the scope of this work.

Secondly, the authors tested whether RsuA plays a role in 30S ribosome biogenesis by investigating if knocking out RsuA affects rRNA maturation. Although the assay shows some evidence of rRNA processing defects in ΔRsuA during early growth stages (4hr and 8hr), I found that this assay is semi-quantitative, and its relationship with streptomycin stress is unclear. What would be the normalization band for the non-denaturing gel assay? Why did the authors not try Northern blot, which is a more canonical assay to test rRNA maturation? The authors should consider testing ΔRsuA cells supplemented with WT or mutant RsuA plasmid in rRNA maturation to figure out the reason for the abnormal growth. Minimally, the authors need to include the raw images for figure 4 in the supplementary files.

Thirdly, how did the author transform data from figure 7a to 7b? Can CD spectroscopy test the fraction of RNA bound with streptomycin? Moreover, using RNase T1 assay to measure streptomycin binding affinity to RNA is an indirect and complicated method, especially considering evidence that streptomycin binds to different locations of modified and unmodified h18. Regardless of the assays used to test binding affinity, I found that the conclusion "Ψ516 modification increases streptomycin binding to helix 18" conflicts with the observation that ΔRsuA cells are more sensitive to streptomycin stress.

Minor comments:

Figure 1: Please annotate the color of each helix highlighted in figure 1b.

Figure 2: It is unclear which line represents WT and which line represents the ΔRsuA cells.

Line 123: Missing citation(s) for catalytically inactive RsuA.

Lines 222-224: Not an intact sentence.

Figure 8: Please include raw images in supplementary files.

Line 284: There is no Sup. Figure 4, should it be Sup. Figure 3?"

Overall easy to read through. 

Round 2

Reviewer 1 Report

The authors have added error bars to their growth rate data, which is greatly appreciated. One minor clarification in the Materials and Methods would be to indicate more clearly whether the statistics were obtained using independent biological replicates.

The authors still have not adequately discussed the implications of their observed effects on G530, which is unfortunate as it is one of the more significant observations in the study. The statement, "...G530, which is known to influence mRNA decoding..." suggests a lack of appreciation of the role of this residue in decoding. They should review and cite  Ogle et al. 2001. They then go on in the revised manuscript to state that this could somehow influence translation rates, but do not state how it would do so. The authors need to explain this further. 

I am still not satisfied with the authors' justification for using the helix 18 model RNAs. I do not suggest that the data obtained with this approach are completely invalid, but the authors all but ignore the suggestion that they include in the text the many caveats associated with using a small RNA to gain insights into ribosome binding to streptomycin. The streptomycin binding site is composed of parts of 16S rRNA helices 18, 27 and 44 and ribosomal protein uS12. Why they would expect streptomycin to bind to these RNAs, given that they cannot possibly form the native conformation in the absence of the pseudoknot, is not clearly justified. The absence of the pseudoknot is especially concerning given that mutations disrupting the pseudoknot are known to confer streptomycin resistance (see Powers and Noller, 1991). Further, aminoglycosides in particular are prone to bind non-specifically to RNA. How confident can the authors be that their effects are specific and not the result of non-specific binding? There are no controls to assess this, and the authors decline to address these issues in the manuscript. 

The authors also continue to ignore the concern about methylation of G527, despite the established role of this modification in streptomycin sensitivity. In their response they state that they do not want to include their own data on G527 methylation, which is fine, but they cannot simply ignore the role of this modification. Given that mutations in 16S rRNA helix 18, helix 27, helix 44 or ribosomal protein uS12 all confer streptomycin resistance, the authors have a serious challenge in arguing that their observations with model stem-loops accurately reflect what is happening in the ribosome.

The quality of English is fine, though a few typos appear.

Reviewer 3 Report

The authors have addressed all my concerns. 

Author Response

Thank you for your time and assistance in improving the manuscript!